# Daytime or Overnight Deliveries? Perceptions of Drivers and Retailers in São Paulo City



**Pedro A. P. Dias [1], Hugo Yoshizaki [2], Patricia Favero [3] and Jose Geraldo Vidal Vieira [4,\*]** 

[1]  Graduate Program in Logistics Systems Engineering, Polytechnical school, São Paulo University, São Paulo 05508-900, Brazil; pedro.parente.dias@gmail.com

[2]  Production Engineering, Poli, São Paulo University, São Paulo 05508-900, Brazil; hugo@usp.br

[3]  Management Engineering Department, Federal University of ABC, Santo André 09210-170, Brazil; patricia.favero@ufabc.edu.br

[4]  Production Engineering, CCGT, Federal University of São Carlos, Sorocaba 18052-780, Brazil

\*  Correspondence: jose-vidal@ufscar.br; Tel.: +55-15-3229-7428

**Abstract:** This research aims to analyze the perception of logistics operators and retailers regarding freight deliveries in the city of São Paulo. Based on a survey applied to 100 logistics operators and 84 retailers, the data were analyzed by descriptive statistics and multiple correspondence analysis (MCA) was used to investigate the logistics efficiency of off-hours deliveries (OHD) and to indicate issues when carrying out OHD. From that sample, noise appears as the most critical issue of OHD for retailers. From the results, most logistics operators and retailers prefer to deliver cargo at night. The advantages of making OHD are the ability to check/store goods, the accuracy in the delivery schedule due to traffic conditions, and the ease of parking a vehicle to offload goods. Public authorities should improve the infrastructure to receive goods, including public lighting conditions and sidewalks. The correspondence analysis method showed that the level of customer service quality depends on the punctuality of the trucks and the ability to check and store goods. Furthermore, by highlighting the logistics efficiency and issues related to daytime and overnight deliveries by carriers and receivers, it can guide public polices and initiatives of other companies, an aspect that has been lacking in the literature.

**Keywords:** overnight delivery; urban freight transportation; retailer; carriers; logistics performance indicator

---

## 1. Introduction

Improving logistic distribution in an urban environment is a complex activity due to the high costs and requirements of good service levels [1]. The intensity of the delivery frequency in light vehicles, environmental issues, and deliveries in unsafe places are factors that hinder urban logistical operations [2,3]. Aiming at greater efficiency in urban logistics, some logistics performance indicators have been adopted by companies that distribute goods in urban centers. Some of these indicators are timeliness of delivery, order errors, incomplete documentation, failure index, and order cycle time [3]. Other indicators are travel time, the number of deliveries in arrears, and the average speed of routes [4]. Managing these indicators allows us to identify where the failures are in the logistics operation of the company, which can lead to the loss of labor productivity, a decline in the level of customer service, and higher costs.

Off-hour deliveries (OHD) are an alternative to urban logistics, increasingly adopted by local governments in large cities [5] concerning the use of existing infrastructure and to enhance the efficiency of goods distribution [6]. However, few cities have implemented OHD initiatives so far [7] due to the

need for acceptance by all stakeholders in urban freight transport [8]. OHD are welcomed by transport operators [9]; in contrast, retailers would prefer to receive goods during regular opening hours as citizens agree that they would prefer not to experience noise during the night and to have their goods available in grocery stores when shopping [10].

Moreover, making OHD causes some relevant issues that should be considered to maximize the efficiency of the deliveries and to avoid externalities [11]. Among these issues, the following can be highlighted: noise emissions [7,12,13]; lack of security for workers, for the cargo, and the establishment [14], and risks associated with public safety. It is worthwhile mentioning that this risk involves cargo theft during the journey, theft during the unloading time, and of course, risking the physical integrity of the employees involved. Cargo theft in Sao Paulo accounts for 47.5% of the Brazilian total [15]. More disturbingly, about 70% of cargo theft results in express kidnapping of drivers. Insecurity has become a major concern for shipping companies, which need to consider the cost of theft and adapt their behavior to reduce risks. Criminals are trying to target certain types of truck or car deliveries based on their type of cargo. The "hot products" that are defined as the riskiest products and preferred by thieves are mostly high-tech products, often electronic products, which can be resold at a high price on the black market. Additionally, based on the statement that public lighting is an important factor in ensuring the safety of the patrimony and society [16], the condition of public lighting can also be an obstacle to off-hour deliveries.

In terms of logistics efficiency, delivering products at night is more efficient [17], as it involves less logistical and operational costs, and it provides more flexibility because it facilitates loading and unloading operations without any traffic congestion [10,18]. However, it is important to analyze the efficiency of the logistics system in detail to indicate what period (daytime or off-hour) is the best choice for logistics operators and retailers, who are directly impacted by OHD solutions since these imply changes in their daily operations [17]. In line with this, some relevant aspects should be considered when comparing OHD to daytime deliveries, such as the route time [4], the organization of the store and the quality of the customer service [17], punctuality of the trucks [19], local traffic congestion, loading/unloading, and parking spaces [2]. By staying less in congested traffic, OHD ensure more predictability and punctuality to logistics routing events, becoming more competitive and reliable [7] given that uncertainties about delivery time should be considered for decision making [20], hence there is less possibility for the presence of customers and other citizens that hinder unloading [17]. Logistics operators prefer assisted OHD to increase their efficiency [7], while retailers incur extra labor costs [6]. The possibility of performing unattended deliveries without having an in-store employee can reduce the store's electricity consumption and the costs of supplementary wages to employees [12,17]. The comparison in the number of fines between off-hour delivery and daytime deliveries is also a fact to be investigated including the efficiency of the two systems [21].

The objective of this article is to analyze the perception of logistics operators and retailers regarding freight deliveries in the city of São Paulo, which is the richest city in Brazil and has a fleet of over 7,500,000 motor vehicles [22], including approximately 200,000 cargo vehicles [23]. In order to improve traffic in large cities, some actions are suggested for cargo transportation, such as installing cross-docking terminals as a way to expedite loading and unloading procedures [24], introducing regulations to restrict truck size [25], and making night deliveries [17]. Furthermore, the main discussion involves investigating problems and solutions to shift part of the urban cargo deliveries to overnight, which can reduce pollutants, a goal that needs a better logistical system [26].

The research questions to answer this objective are: (a) What are the main issues arising from OHD? (b) Are OHD perceived as more efficient than daytime deliveries at an operational level, by the logistics operators and retailers? Concerning the first question, the following aspects will be investigated: security during the reception of goods at night, the noise emitted while receiving OHD and public street lighting around the stores. For the second question, the following aspects will be verified: route time, time required to find a loading and unloading bay, type of fine, level of stress (drivers); peak time of customers moving through the store, the best time to allocate goods on the shelf, queue of

trucks (traffic congestion), punctuality of the trucks, and ability to check and stock goods [12,16,17]. Moreover, through interviews with drivers and retailers, this work investigates qualitative aspects that negatively influence overnight delivery operations [14,27].

As São Paulo is among one of the first municipalities in developing countries to test OHD as a transport policy [28], this work is relevant as it shows the acceptance of this urban distribution policy from retailers and logistics operators. Analyzing the proposed aspects may corroborate whether, or not, the urban logistic solution of using OHD in the megacity of São Paulo is a feasible solution that can be applied to the rest of the city. The focus of local regulations and any freight distribution system is to ensure equality between social well-being, costs, and benefits for most of the stakeholders involved. It is also the first scientific survey to explore off-hour deliveries in the megacity of São Paulo.

The structure of this paper is as follows: Section 2 describes the materials and methods. Section 3 shows the results based on the perspectives of drivers and receivers. Section 4 discusses the results based on the theoretical framework, and finally, we provide the overall conclusions, future works, and limitations of the study in Section 5.

## 2. Materials and Methods

The research methodology is based on survey data of retail companies that participate in off-hour deliveries through empirical research.

### 2.1. Pilot Test

The initiative to develop an OHD pilot started in January 2014, when the São Paulo Secretary of Transport organized a technical group to evaluate the feasibility and plan for this pilot project in São Paulo. The task group included: the Department of Road System Operations (Departamento de Operações do Sistema Viário—DSV); the Traffic Engineering Company (Companhia de Engenharia de Tráfego—CET), São Paulo Motor Carrier Syndicate (Sindicato de Empresas de Transporte de Carga de São Paulo e Região—SETCESP); the Retail Development Institute (Instituto de Desenvolvimento do Varejo—IDV), an association of the top retailers in Brazil, and the Center for Logistics Systems Innovation at the University of São Paulo (CISLOG/USP). The main issues to be evaluated were noise, safety, traffic, and costs/productivity of night operations. To better understand OHD and control the pilot evaluation, an 11 km$^2$ area, basically within the Maximum Circulation Restriction Zone (Zona de Máxima Restrição de Circulação—ZMRC), was chosen by its characteristics of mixed land use and low–average safety risk. It bordered a very important ring road and encompassed major traffic generators, such as large stores, shopping malls, as well as traditional street establishments, residential areas, few industries, and service locations, mostly in middle-class neighborhoods. After several workshops with the stakeholders, including companies (retailers and logistics operators), in October 2014, the pilot study began with 45 different establishments, including eleven companies: two large supermarket/grocery retail chains (one of them is the top Brazilian retail group), two top pharmacy retails chains, two top department store groups, one in general merchandise and the other in the fashion business (shopping mall anchor stores), a top cosmetics/beauty manufacturer (with both private and franchised stores), a paper bag manufacturer (supplier to high end stores), a fast-food chain, a top home improvement materials superstore chain, and a large beverage industry. Three companies already had experience with night deliveries: both grocers (with less than 10% of deliveries within the ZMRC made at night, usually to 24/7 stores and to small, neighborhood store formats), and one of the pharmacies, which at that time had more than 30% of stores supplied at night in the São Paulo metropolitan region (SPMR) [28].

The findings of this pilot test have shown that there were no registered complaints of noise, and no safety and security occurrences. Regarding operational and economic perspectives, average travel speeds were higher, and effective productivity gains could be achieved in some operations.

### 2.2. São Paulo City: Restrictions and Regulations

The aim of this topic is to present the main public policies for urban transport systems adopted in the city of São Paulo that may impact the cargo distribution strategy of the companies involved. Over the last decades, the São Paulo city government, through the CET has been adopting measures that have restricted the movement of trucks in the city and stimulated night deliveries. Urban logistics solutions focus on restrictive measures, and this is also the case for São Paulo city [29]. Easy to implement and not costly, more generally, regulations seem to be a quick answer to social and environmental issues that freight transport brings into cities [30].

The OHD system in São Paulo city allows vehicles to deliver goods between 10 pm and 6 am. On the other hand, there is a possibility of delivering goods using smaller trucks, the Urban Cargo Vehicle (Veículo Urbano de Carga—VUC) during the day and on certain routes. VUCs need to meet the following specifications: maximum height of 2.20 m and the maximum length of 7.20 m for vehicles, which were produced from 2015 onwards.

In 1997, Operation Pico Time was implemented, popularly known as "rodizio," to minimize the emission of pollutants. In 2008, the city of São Paulo also decided to restrict the movement of trucks, as a way to minimize traffic in the city. Rotating vehicles restrict the movement of cars and trucks according to the reconciliation between the end of the license plate and the day of the week (see Table 1). Trucks cannot circulate during rush hours (i.e., from 7:00 a.m. to 10:00 p.m. and 5:00 p.m. to 8:00 p.m.). This "license-plate-based car rotation scheme" may influence the decision to deliver at night.

**Table 1.** Last digits of the license plate and restriction per day.

| Day of the Week | Monday | Tuesday | Wednesday | Thursday | Friday |
|---|---|---|---|---|---|
| License digits of license plate | 0 and 1 | 2 and 3 | 4 and 5 | 6 and 7 | 8 and 9 |

Source: CETSP (2019).

For example, vehicles with a license plate ending in 0 or 1 are prohibited from driving on Mondays from 7:00 a.m. to 10:00 p.m. and 5:00 p.m. to 8:00 p.m., except holidays, in the expanded center-zoning that is delimited by the mini-ring road. However, this restriction does not apply when transporting perishable food and essential public services, such as garbage collection, fire, and postal services.

In addition to the rotation, since 1982, the government of São Paulo has been adopting a series of measures as a way to mitigate the inconveniences caused by the traffic of trucks in the city. Measures restricting truck traffic in urban centers are some of the most commonly used measures in emerging countries. However, they do generate a great deal of debate about their effectiveness in reducing congestion, pollution, and the number of accidents involving cargo vehicles as logistic service providers may use a larger number of smaller and exempt cargo vehicles of traffic restriction to meet the daily demand and the level of service demanded by the customer [31].

The following will describe the current public policies that regulate the movement of cargo vehicles in certain perimeters and ways of the city.

ZMRC—Maximum Circulation Restriction Zone: This was established so that the movement of the urban load occurs, preferably at dawn and at night. Transit of trucks is forbidden from Monday to Friday from 5 am to 9 pm, and Saturdays from 10 am to 2 pm, except holidays. Vehicles with special authorization transporting perishable food products are allowed to run between 5:00 a.m. and 12:00 a.m. VUCs are allowed to run full time.

VER—Restricted Structural Routes— are roads, tunnels, viaducts, and high-traffic bridges responsible for the articulation of the road system between regions of the city and also accessed for long-distance travel. They have local regulations regarding circulation times for vehicles, such as VUCs and the transportation of perishable foodstuffs.

Vehicles, trucks, and VUCs, whose activities or nature of transported products are classified as exceptions provided for in the ZMRC and VER regulations, must hold a Truck Card, which is

a special permit for movement and parking obtained by registering at the Municipal Secretariat of Transportation (SMT in Portuguese). For example, VUCs have specific times and days for circulation in these zones and restraint routes for activities, such as garbage collection, transportation of perishable foods, and transportation of values.

ZERC—Circulation Restriction Special Zone—residential area in which truck traffic is prohibited, including for VUCs.

In addition to the traffic restriction policies adopted by the São Paulo government, it is also worth noting Federal Law No. 12,619, which regulates the working day of the driver ensuring a working day with a minimum interval of 1 h for meals, 71 daily rest intervals of 11 h every 24 h and a weekly rest of 35 h. This measure became known as the "driver's law," and there is a concern about the impact it may have on urban freight distribution.

### 2.3. Sampling Technique

Following the previous results and based on the literature, a questionnaire was prepared and distributed to 84 receivers (representatives of the retailers) from October 2014 to March 2016 through face-to-face interviews; some of them participated previously in the pilot test. These 84 stores belonged to 17 companies (four came from the pilot study) in the retail sector, among them: grocers, pharmacies, general merchandise, civil engineering material, office stationery, and household articles. Around 45% of these companies are in the top five of their industrial sector. To better understand the operators' perceptions on OHD, a questionnaire was prepared and conducted face-to-face with 100 drivers (representatives of the carriers) that make or have made off-hour deliveries to these retailers in that region. The sample of drivers was random due to the difficulty to estimate the real number of drivers and carriers. Although it is impossible to induce the entire population, the sample can be considered representative of what the main proposal is.

### 2.4. Data Analysis

Descriptive statistics were prepared with frequency analysis and some measures of position and dispersion, such as mean, median, and standard deviation. The multivariate statistical method was also adopted to obtain additional information. Multiple correspondence analysis (MCA) was used on two occasions: to identify the most critical variables of efficiency in OHD and to indicate the major problem in carrying out the OHD.

MCA can be defined as a multivariate technique that can represent a summary of the relations existing between a set of categorical variables forming a multiple contingency table. If we have only two variables in a contingency table, we use correspondence analysis (CA); if the variables are more than two, we use MCA [32].

MCA is a dimensionality reduction factor technique in the same family as the most popular principal component analysis (PCA), but specifically designed to deal with nominal or ordinal categorical data. MCA deals specifically with categorical data, assigning scale values to the categories of the variables and maximizing the variance of those scores to find: (1) the associations between the variables, and (2) the proximity between individuals. Analogous to PCA, MCA provides eigenvalues and factor loadings, but its emphasis is on the geometric representation of data structures. Objects (respondents) and variables (variable categories) are represented as points in a weighted Euclidean space with very few dimensions (often only two or three) so that spatial proximity of categories or individuals indicates similarity and vice versa [33–35].

In this research, this technique shows, in a graph, the perception map regarding some attributes related to the two types of delivery (daytime or nighttime).

### 2.4.1. Definition of the Category Coordinates (Scores) in the Perceptual Map

Following the steps proposed by [36], the coordinates or scores in a multidimensional space can be defined.

Imagine a database with two categorical variables, where the first has $I$ categories and the second $J$ categories. From this database, it is possible to define a cross-tabulation or contingency table which shows the observed absolute frequencies of the categories of the two variables, where given cell $ij$ contains a certain amount $n_{ij}$ ($i = 1, \dots, I$ and $j = 1, \dots, J$) of observations. A total number of observations $N$ in the database can, therefore, be expressed by:

$$N = \sum_{i=1}^{I} \sum_{j=1}^{J} n_{ij} \tag{1}$$

The general representation of the contingency table, including the total values of the observed absolute frequencies in each row and column, is in Table 2.

**Table 2.** Contingency table with total values by row and column.

|       | 1 | 2 | ... | $J$ | Total |
|-------|-----------|-----------|-----|-----------|-------------|
| 1 | $n_{11}$ | $n_{12}$ | | $n_{1J}$ | $\sum r_1$ |
| 2 | $n_{21}$ | $n_{22}$ | ... | $n_{2J}$ | $\sum r_2$ |
| ... | ... | ... | | ... | ... |
| $I$ | $n_{I1}$ | $n_{I2}$ | | $n_{IJ}$ | $\sum r_I$ |
| Total | $\sum c_1$ | $\sum c_2$ | ... | $\sum c_J$ | $N$ |

where:

$$\sum c_1 + \sum c_2 + \dots + \sum c_J = \sum r_1 + \sum r_2 + \dots + \sum r_I = N \tag{2}$$

From a contingency table with dimensions (I × J), we can define the mass concept which represents a measure of influence or preponderance of some category compared to the others, based on its observed frequency.

Based on the average mass values in row and column, we can define two diagonal matrices, $D_r$ and $D_c$, containing these respective values in their main diagonals. Therefore, we have:

$$D_r = \begin{pmatrix} \dfrac{\sum r_1}{N} & 0 & \cdots & 0 \\ 0 & \dfrac{\sum r_2}{N} & \cdots & 0 \\ \vdots & \vdots & \ddots & \vdots \\ 0 & 0 & \cdots & \dfrac{\sum r_I}{N} \end{pmatrix} \tag{3}$$

and

$$D_c = \begin{pmatrix} \dfrac{\sum c_1}{N} & 0 & \cdots & 0 \\ 0 & \dfrac{\sum c_2}{N} & \cdots & 0 \\ \vdots & \vdots & \ddots & \vdots \\ 0 & 0 & \cdots & \dfrac{\sum c_J}{N} \end{pmatrix} \tag{4}$$

To make the perceptual map, we need to calculate the eigenvalues ($\lambda^2$) of a matrix $W$ defined as:

$$W = \begin{pmatrix} w_{11} & w_{12} & \cdots & w_{1J} \\ w_{21} & w_{22} & \cdots & w_{2J} \\ \vdots & \vdots & \ddots & \vdots \\ w_{I1} & w_{I2} & \cdots & w_{IJ} \end{pmatrix} \tag{5}$$

The diagonal matrix of eigenvalues of the matrix $W$, called $\Lambda^2$, is expressed by:

$$\Lambda^2 = \begin{pmatrix} \lambda_1^2 & 0 & \cdots & 0 \\ 0 & \lambda_2^2 & \cdots & 0 \\ \vdots & \vdots & \ddots & \vdots \\ 0 & 0 & \cdots & \lambda_m^2 \end{pmatrix} \tag{6}$$

where each $\lambda_k^2$ corresponds to the partial principal inertia of the kth dimension, and $\lambda_k$ to its singular value. Thus, having defined the eigenvalues of the matrix $W$, we can obtain the eigenvectors of the same matrix, which we will call:

$$V = \begin{pmatrix} v_1 \\ \vdots \\ v_J \end{pmatrix} \tag{7}$$

and

$$U = \begin{pmatrix} u_1 \\ \vdots \\ u_I \end{pmatrix} \tag{8}$$

Defining the diagonal matrix of eigenvalues $\Lambda^2$ and the eigenvectors $U$ and $V$, the coordinates (abscissa and ordinate) for the perceptual map elaboration can be calculated based on the following expressions:

$$X = D_r^{-1}.\left(D_r^{1/2}.U\right).\Lambda \tag{9}$$

$$Y = D_c^{-1}.\left(D_c^{1/2}.V\right).\Lambda \tag{10}$$

The $X$ and $Y$ coordinates obtained through expressions (9) and (10) are used to elaborate a perceptual map known as the symmetric map, where the points representing the rows and columns of the variable categories have the same scale (symmetric normalization).

## 3. Results

### 3.1. The Perspective of the Drivers: Respondent Profile and Characterization of Deliveries

About 50% of drivers have more than ten years of experience of driving; 70% stated that they preferred to work at night, and 80% feel higher stress levels while working during the day. Stress is caused by different means: noise, mechanical conditions of the vehicle, vibrations, repetitive efforts, posture while driving, and lack of safety during unloading.

Table 3 shows the average route time and the average time for finding parking bays, both in the daytime/overnight period.

**Table 3.** Route time and to find a place to unload during the daytime.

| | Route Time (Hours/Day) | | Finding Parking Bays (min) | |
|---|---|---|---|---|
| Measurements | Daytime | Overnight | Daytime | Overnight |
| Average | 9.15 | 4.4 | 47.83 | 11.82 |
| Median | 9.5 | 4 | 40 | 15 |
| Standard deviation | 3.19 | 2.02 | 35.73 | 10.31 |

The timeliness of urban deliveries is also different when deliveries occur in the daytime or night period, as shorter route times in the night period reflect greater efficiency in logistic operations [12] and more reliability regarding the timing of the trucks [27]. For 71% of the drivers, urban deliveries are

more punctual if they are done in the night period. Usually, these deliveries are performed by trucks and small 3.5-ton trucks, i.e., VUCs that can circulate anytime during the day through the ZMCR.

Regarding the fines, from the 100 drivers interviewed, 74 reported having incurred fines; these fines were imposed mainly because of drivers parking in a prohibited parking space (45) by restriction to truck traffic (29) in ZMRC. It is worth mentioning that at night, there are no fines for traffic restriction; this rule prohibits larger trucks from circulating between 5 h and 20 h. Although we have not asked the drivers about the possible influence of a license-plate-based car rotation scheme, the literature shows that, because of this scheme, logistics service providers use a large number of light vehicles for daily deliveries to a large number of stores, which contributes to many urban freight transport issues (traffic congestion, long load–unload queue, intensification of environmental problems) while carriers do not see it as a problem [2], maybe because they have been planning their vehicles according to this schedule.

Characterization of the Actions of Public Power and the Structure of the Reception

About 46 drivers perceived that the public lighting is satisfactory, 10 considered it regular, and 26 understood that the conditions of this structure of analysis were not satisfactory. The problems concerning parking during the day are unanimous among the drivers because everyone who answered the question about demarcating loading and unloading positions said that this action of the public power is very important to expedite unloading. Not only demarcating the loading and unloading positions but also inspecting them and penalizing the vehicles that unduly park in them. The drivers also reported that holes and unevenness in sidewalks make working with pallet trucks difficult, or even manually loading boxes with goods. Among the damage caused during the unloading process due to the poor condition of sidewalks, drivers highlighted the following problems: damage to goods, damage to pallets, and risk of accidents, among others. The need to improve sidewalks is reflected by the unanimous response of the drivers as 57 drivers consider it very important to improve the sidewalks.

Of the 57 drivers, only 13 of them (23%) prefer to make deliveries without any help from the recipients' employees. Drivers who responded favorably to unattended deliveries reported that this type of delivery is more responsive and improves the logistic performance of the delivery schedule.

It is a fact that there is a lack of structure for receiving goods. Among the main problems is the fact that there are too many steps, narrow doors, a queue of trucks, and inadequate conditions of the receiving docks [25]. Too many steps on the sidewalks have been cited as the reason for delays in delivery and also causing physical fatigue to drivers and assistants due to the extra effort exerted. It should be mentioned that out of 57 drivers, 52 (91.2%) reported frequent deliveries to stores with too many steps in unloading and storage areas of products within the stores. Narrow doors make it difficult to carry boxes or pallets through, increasing the unloading time, and in the case of overnight deliveries, unloading is expected to be more agile to avoid the risk of burglary. The long queues of trucks, while waiting to unload merchandise, are also cited by drivers as a favorable reason for delivery freight at night. The drivers declared that there was a lack of adequacy between the receiving docks and the type of trucks at the time of delivery. When the property has a raised dock, the truck ramp does not need to be used. Moreover, with the raised dock, the engine of the truck does not have to keep running during unloading to move the ramp, which can avoid noise problems and also reduce the emission of pollutants.

From the 74 drivers, 55 (65%) reported having problems with noise during the unloading. About 89% of drivers declared insecurity during the night deliveries. However, even with the feeling of insecurity, 70% of the drivers said they prefer to work at night, which is compatible with previous results [16].

*3.2. Receivers' Perspective: Characterization of Public Power Actions and Reception Structure*

From the sample of receivers, at least 50% of them have 3 years in the position, a short time compared to the case of drivers. From 83 respondents, only 26 (31.3%) rated public illumination as

very bad or bad; 44 (57.1%) respondents agreed with the delimitation of loading/unloading in the vicinity, and most retailers did not have an internal dock. It is worth remembering that in some cases the dock only has the capacity for a truck, which forces the other cargo vehicles to park on the street, with the possibility of not finding parking spaces, parking in a prohibited place, receiving a fine, and most importantly the risk of being mugged. Out of 80 respondents, 74% stated that it is very important for the public power to improve the sidewalks. In many cases, unloading is done on sidewalks, thus improving this structure could expedite the delivery schedule and contribute to improvements in urban logistics [30].

From the 84 establishments, only 17.8% make non-assisted deliveries. These are linked to low value-added products, such as fruit, vegetables, and greens. As with drivers, the queue of trucks is the most recurrent problem in unloading operations, and only 20% of receivers perceive structural problems as steps and a narrow door for receiving goods.

It can be observed that 63% of the respondents consider overnight deliveries to be more punctual. OHD are more assertive, contributing to the efficiency of urban logistics [16]. About 58.3% of respondents said nighttime unloading activities are more agile, which makes it easier to check and organize stores because of the absence of customers. However, overnight deliveries increase the risk of fatigue, insecurity, and difficulty in adapting the human body to nighttime schedules. Moreover, 58% of respondents stated that there is no security in receiving cargo at night.

It should be mentioned that 39.2% of those interviewed reported having problems receiving complaints from the neighbors during overnight unloading. Some receivers pointed out that lighter goods, such as bread, fruit, and vegetables, make less noise during unloading.

An interesting fact is that receivers maintain the same inventory and transportation policy by using larger vehicles to ensure more efficiency at night by increasing the order size and consequently decreasing the frequency of deliveries. Night deliveries are preferred by 65% of respondents compared to daytime deliveries.

*3.3. Association between Delivery Time Preference and Problems*

The multiple correspondence analysis was applied to highlight the main problems arising from carrying out the type of deliveries, according to drivers (Figure 1) and receivers (Figure 2). Thus, it was observed which categories are closer to the daytime delivery option, and which are closer to night deliveries for both investigated subjects. The variables in the analysis related to night delivery problems are noise, public safety, and lighting. Before applying MCA, the chi-square test was used to verify if there was an association between the categories of the variables. The null hypothesis of the test was rejected, indicating that there is an association between the categories of the variables (i.e., type of delivery, lighting, noise, and public safety).

The models were implemented in the SPSS software. According to the equations presented in Section 2.4.1., the coordinates (scores) of the variable categories were calculated. Table 4 presents the coordinates of the categories of each variable (delivery time preference and problems, according to drivers) for two dimensions. Similarly, Table 5 presents the coordinates of the categories of each variable (delivery time and problems, according to receivers) for two dimensions.

Based on these scores that represent physical distances between the variable categories, we then generated the perceptual map (Figures 1 and 2), so that the greater the distance between them, the smaller the association is.

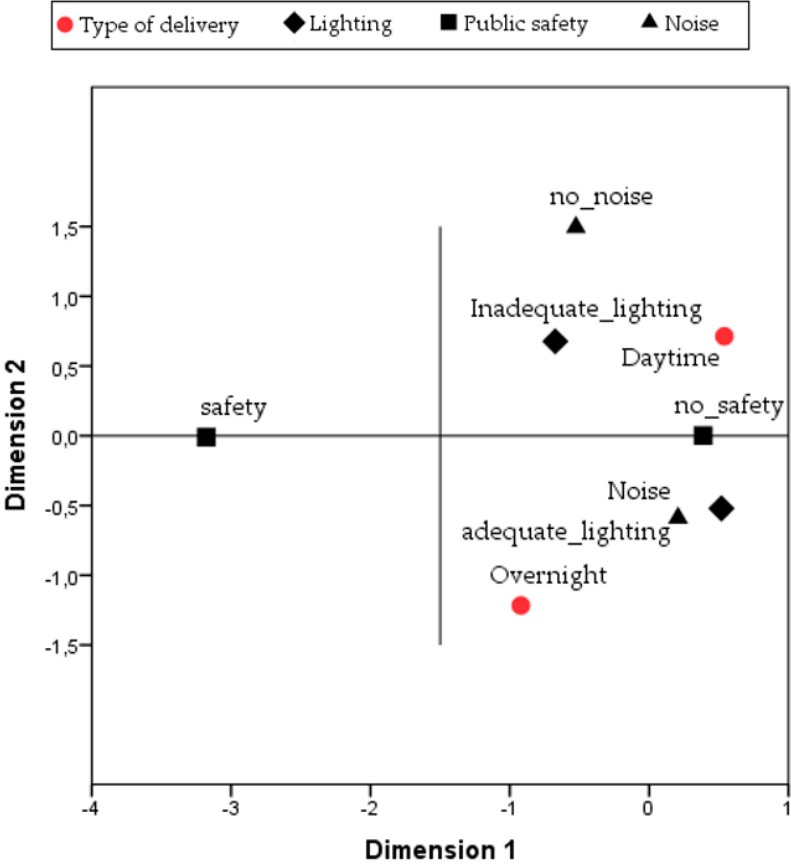

**Figure 1.** Association between delivery time preference and problems according to the drivers.

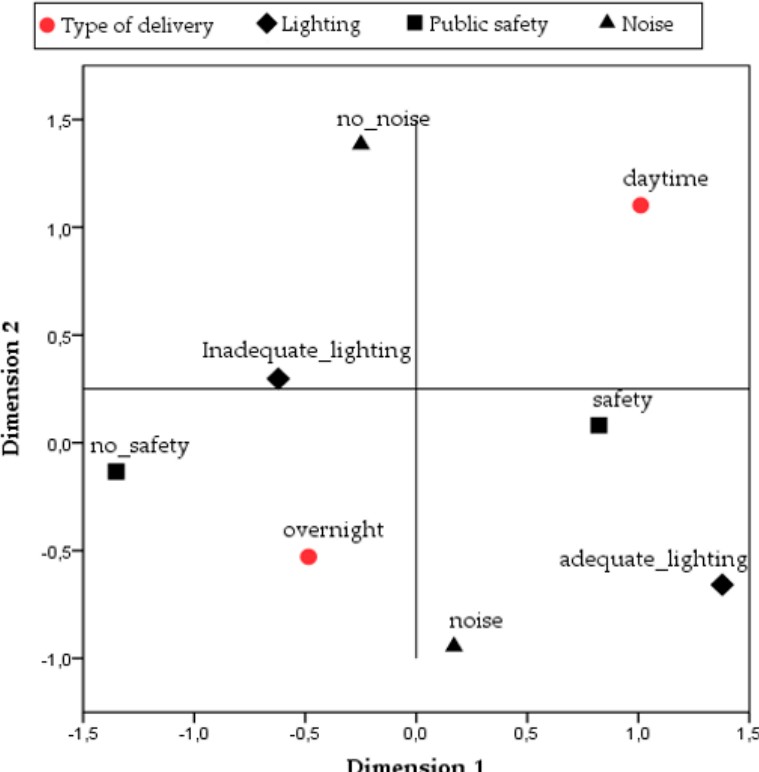

**Figure 2.** Association between delivery time and problems according to the receivers.

**Table 4.** Coordinates of the categories of each variable (delivery time and problems, according to drivers).

| Variable | Category | Frequency | Centroid Coordinates | |
|---|---|---|---|---|
| | | | Dimension 1 | Dimension 2 |
| Type of delivery | daytime | 29 | 0.540 | 0.713 |
| | overnight | 17 | −0.921 | −1.217 |
| Lighting | inadequate_lighting | 20 | −0.674 | 0.677 |
| | adequate_lighting | 26 | 0.518 | −0.521 |
| Public safety | no-safety | 41 | 0.387 | 0.001 |
| | safety | 5 | −3.177 | −0.010 |
| Noise | noise | 33 | 0.207 | −0.589 |
| | no-noise | 13 | −0.526 | 1.495 |

**Table 5.** Coordinates of the categories of each variable (delivery time and problems, according to receivers).

| Variable | Category | Frequency | Centroid Coordinates | |
|---|---|---|---|---|
| | | | Dimension 1 | Dimension 2 |
| Type of delivery | daytime | 24 | 1.011 | 1.102 |
| | overnight | 50 | −0.485 | −0.529 |
| Public safety | safety | 46 | 0.822 | 0.081 |
| | no-safety | 28 | −1.351 | −0.134 |
| Noise | no-noise | 30 | −0.249 | 1.385 |
| | noise | 44 | 0.170 | −0.944 |
| Lighting | adequate_lighting | 23 | 1.378 | −0.659 |
| | inadequate_lighting | 51 | −0.621 | 0.297 |

For drivers (see Figure 1), daytime delivery is closer to two categories: no public safety ("no_safety") and inadequate lighting ("inadequate_lighting"). In other words, for drivers, there is a need to create conditions to ensure that daytime unloading occurs safely. Moreover, there is a probability of noise during overnight deliveries. For receivers (see Figure 2), overnight deliveries are close to the "Noise" category (noise problems) and "no_safety" category.

*3.4. Association between Delivery Time Preference and Logistics Efficiency*

It was verified which variables contribute the most in improving the efficiency of urban deliveries per period. The following variables were used: punctuality, agility to check and to store, quality of customer service, parking spaces, and truck queue length. The chi-square test was also applied before MCA to verify if there was an association between the categories of these variables. The null hypothesis of the test was rejected, indicating that there is an association between the categories of the variables. Table 6 presents the coordinates (scores) of the categories of each variable for two dimensions. Based on these scores, we then generated the perceptual map (Figure 3) which shows an association between the preferences for daytime deliveries with three logistics efficiency variables: agility to check and to store goods, quality of customer service, and punctuality.

**Table 6.** Coordinates of the categories of each variable (deliveries and logistics efficiency, according to receivers).

| Variable | Category | Frequency | Centroid Coordinates | |
| --- | --- | --- | --- | --- |
| | | | Dimension 1 | Dimension 2 |
| Type of delivery | daytime | 20 | 1.390 | 0.575 |
| | overnight | 42 | −0.662 | −0.274 |
| Punctuality | less_punctuality | 23 | 1.147 | −0.360 |
| | moss_punctuality | 39 | −0.676 | 0.212 |
| Agility to check and to store | less_agi | 21 | 1.498 | −0.071 |
| | more_agi | 41 | −0.767 | 0.036 |
| Quality of customer service | less_quality_CS | 16 | 1.746 | −0.042 |
| | more_quality_CS | 46 | −0.607 | 0.015 |
| Parking spaces | parking-spaces | 41 | −0.146 | 0.814 |
| | no-parking-spaces | 21 | 0.284 | −1.590 |
| Queue of trucks | no_queue | 35 | −0.142 | −0.946 |
| | queue | 27 | 0.184 | 1.227 |

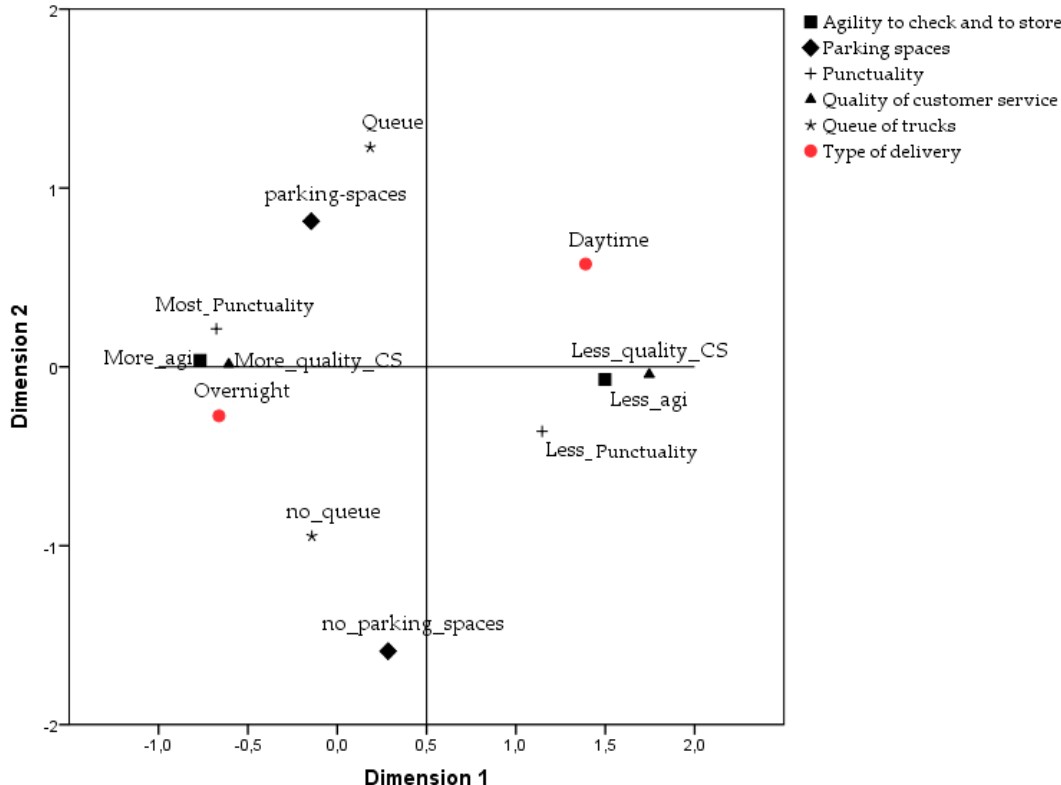

**Figure 3.** Association between deliveries and logistics efficiency according to the receivers.

For receivers who prefer overnight deliveries, there are three categories of service that are closer to each other: quality of customer service with overnight pick-up ("More_quali_CS"), agility in checking and storing overnight ("More_agi"), and more assertive punctuality during the night ("Most_punctuality"). The closer positioning reveals that night delivery can improve the quality of customer service due to greater assertiveness regarding the arrival time of the truck and the greater agility in checking and storing goods [16]. The agility in checking and storing merchandise at night time may be better as the stores are empty, and employees have more space for the boxes and more peace of mind to store the goods on the shelves.

Another interesting finding is that the main advantage of night deliveries is not associated with the unloading conditions (queue of trucks and parking spaces), but rather with the internal organization of the store, from greater certainty in the truck time arrival to the agility to check and store the merchandise.

## 4. Discussion

In general, the literature shows that OHD can improve traffic congestion and travel times for road users during daytime hours; reduce impacts related to environmental issues (noise, accidents, and emissions), increase competitiveness for logistics operators (e.g., less delivery traffic conflicting with customers and availability for unloading and parking spaces), increase delivery reliability for receivers, provide more flexible services, increase safety by reducing the conflicts between trucks, passenger cars, cyclers, and pedestrians [6,7,16].

Night deliveries in São Paulo are preferably assisted [7]. It was observed that each delivery consisted of a driver and two assistants. In some cases, the presence of an armed escort comprising two security guards in a passenger car. The most suitable products to be delivered at night are General Merchandise, perishable food (e.g., fruit, vegetables, and greens), and bread as they have low added value, the unloading occurs more quietly, they are lighter products, and are not risky loads (low value). Groceries can be unloaded between the hours of 9:00 pm and 5:00 p.m.

Therefore, the risks associated with public safety (i.e., lack of security for workers, for the cargo and the establishment) are a problem for drivers and receivers, and it is regardless of the delivery time. Such security problems may generate extra costs for companies (e.g., labor costs, escort vehicles, replacement of stolen products and vehicles, and so on) [2], and these problems can help decrease the livability and attractiveness of the city [6], mainly when OHD take place.

The comparison of daytime deliveries and off-hour deliveries indicates, from the drivers' point of view, the reduction of time searching for a loading and unloading bay in the central region, which seems to have a more significant impact if urban deliveries are performed at night, and this factor can reduce the route time and improve the punctuality of the trucks. Moreover, OHD may reduce the number of trucks waiting for unloading, which decreases the waiting time and total route time. These advantages seem to outweigh the feeling of insecurity and noise issues, as most drivers prefer OHD. It is worth remembering that noise is the main obstacle to overnight deliveries [13].

From the viewpoint of the receivers, the possibility of improving the quality of customer service is the most attractive factor in carrying out OHD. Greater certainty about truck arrival times allows deliveries to be more controllable and more punctual during the night. The result of multiple correspondence analysis also provides a better understanding of the advantages of OHD for retailers; it was a gap in the literature [7]. Moreover, the absence of customers and other routine nuisances offers more agility to the employees who check and store goods. These aspects contribute to the quality of customer service. They are preferred by receivers (especially when deliveries are outside business hours), despite concerns about unsafe conditions and the possibility of conflicts between drivers and residents over the noise caused by deliveries.

## 5. Conclusions

We conclude that, from the standpoint of the direct stakeholders, the advantage of making overnight deliveries concerns the ability to check and store the goods, the reliability on the delivery schedule due to better traffic conditions, the facility to park the vehicle to perform unloading activities, and the reduced route time needed [5].

Given that most logistics operators and retailers prefer to continue to perform the overnight unloading of goods, public authorities should improve public lighting conditions and inspect loading/unloading places in locations close to bars and restaurants acting at night. In line with this, a policy for OHD in the city of São Paulo presupposes taking effective measures to mitigate noise (for instance, adopting low-noise technology [6]), improving the infrastructure and public safety

to receive merchandise and increasing the number of places for loading and unloading. Structural investments are also needed both by parts of transport companies and by retailers, as highlighted in the literature. There were structural failures of the receiver's responsibility, such as too many steps and narrow doors; a lack of public lighting, which is the responsibility of the public power, and finally, poor road conditions and lack of parking spaces as a critical problem of the responsibility of the receivers and the public power [37]. Along these lines, OHD with an urban distribution center [7], better planning of cargo deliveries (during the day and also on weekdays) could be an alternative. For example, there are low-income neighborhoods where shopkeepers receive goods preferentially on Thursdays. Therefore, better delivery and receiving planning, as well as changes in road control, could be changed to facilitate the flow of cargo vehicles.

As our study focuses on the perspective of retailers and logistics operators, there is no discussion regarding the impressive potential for pollution reduction from OHD in large, congested cities, as it is well explored elsewhere [5].

It should be mentioned that the São Paulo OHD Pilot Test was an acknowledged success, both by the city government and private companies [28]. The mayor of São Paulo stated that he considers "freight transport as public transportation because it takes care of public welfare. Trucks supply commerce and manufacturing and generate jobs," in August 2015. That mayor installed a Division of Freight Transport in the traffic authority to deal with all aspects regarding freight transportation in the city, and night deliveries were instituted as a formal policy of freight mobility; it must be noted that the new mayor that took office in 2017 kept both the new division and the night deliveries policy. For the private sector, all companies that participated in the OHD Pilot and have a verticalized supply chain greatly expanded night deliveries, most of them finding a balance of about 40% of OHD; today, more new large retailers also started OHD after the pilot, using material developed from that case and available freely on the internet.

Other critical factors could be considered in the analysis of logistics efficiency as a way to measure service levels of night and day deliveries. For example, loading/unloading time, product quality, losses, theft, extra labor costs. In line with this, new research could consider these and other important issues, such as congestion, accidents, the severity of accidents, and deterioration of products transported. In addition, some measures for reducing the environmental effects of freight transport could be considered at the urban level [38].

As future work, specific areas inside São Paulo city (population zone with high/low income, or area with different density) could be chosen to develop OHD using simulation models of the distribution system [39]. Because the data were collected in a not too recent period, a new survey may show new insights and implications for all stakeholders.

**Author Contributions:** P.A.P.D. contributed to the data acquisition, conducted the questionnaire, drafted the paper, and performed statistical analysis. H.Y. contributed by reviewing and editing the paper, conceptualization, and project administration. P.F. contributed to the statistical analysis, methodology, and supervision of data analysis. J.G.V.V. contributed to the writing and editing of the paper, methodology, conducted the questionnaire, drafted the original paper, performed statistical analysis, formal analysis, and supervision.

**Funding:** This research was funded by the Brazilian Coordination for the Improvement of Higher Education Personnel (CAPES), the National Council for Scientific and Technological Development (CNPq), grant numbers 309516/2016-1 and 311021/2016-6, and the Fundação de Amparo à Pesquisa do Estado de São Paulo—FAPESP, grant number 2017/06074-7.

**Conflicts of Interest:** The authors declare no conflict of interest.

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
