# Peer review of "Daytime or Overnight Deliveries? Perceptions of Drivers and Retailers in São Paulo City"

_sustainability, doi:10.3390/su11226316_

Round 1

Reviewer 1 Report

The paper promise to be very interesting from the point of view of urban city and scheduling problems. The second problem might be a subject matter to future research. Authors are encourage to do so. Now, the paper need some additional elaboration.

Data for the research are relatively old (2014-2015). Authors should make some statement about that fact. Same parts of the paper are compared to 2010 which is also quite obsolate data.

Some companies names are given and another are not. Authors should make some statement about that as well.

Table 1 should be mentioned more - what is its reason and results, any objectives? It is not clear.

Incorrect recording of hours should be improved: ”to run between 5 and 12h.”

Acronyms such FLV and other should be described with its FIRST use.

”We then generated scores that represent physical distances on the perceptual map” - how were these prepared? Authors should describe the methodology and give results not in figures’ way only. These should be taken into consideration for figures 1-3.

The kind of challenge mentioned in the paper is described in qualitative format. It needs quantitative one, therefore it is suggested to continue the research as such. Some method as FMEA, prediction markets, simulation modelling might be of sophisticated use.

Why is this reference at the end of the sentence? What is its meaning: ”As future work, specific areas inside São Paulo city could be chosen to develop OHD, taking account the opinion of main stakeholders: retailers, logistics operators, residents, and public administrations [27].” How does it support the meaning?

Since „The problems in parking during the day are unanimous among the drivers” it might be good to co-operate on this matter for future research as e.g. in  especially Kostrzewski M., Varjan P., 2018, The Issue of Parking Areas Conditions in Surrounding of Logistics and Production Facilities in Slovakia and Poland, Proceedings of 22nd International Scientific Conference. Transport Means 2018, Part II, October 03 – 05, 2018, Trakai, Lithuania, pp. 791-796 or Igarashia K., Okumura S., https://doi.org/10.1016/0886-7798(93)90138--L

Prediction markets might be used as well, based on Czwajda et al., http://doi.org/10.17270/J.LOG.2019.329

”It is a fact that there is a lack of structure for receiving goods, among the main problems are the excess of steps, narrow doors, the queue of trucks and the inadequate conditions of the receiving docks.” -  yes that’s right, therefore methods for facilities design might be used for future research. By the way, it is highly proposed not to use such generalised statements without any references.

Authors stated ”The long queues of trucks while waiting to unload merchandise is also cited as a favorable reason for delivery freight at night” however no citation is given. It should be specified.

Future results should be mentioned in the paper in more developed way. Aspects of urban logistics might be enriched with simulation studies as it was suggested before, e.g. proposed in: https://doi.org/10.1016/j.proeng.2017.06.077

Author Response

Dear Reviewer,

Thank you for reviewing our article. Your comments have enabled us to improve our paper. The amended document accompanies this response. Please see our responses below for specific details on how we followed your comments and suggestions.

# Reviewer 1

The paper promise to be very interesting from the point of view of urban city and scheduling problems. The second problem might be a subject matter to future research. Authors are encourage to do so. Now, the paper need some additional elaboration.

 1) Data for the research are relatively old (2014-2015). Authors should make some statement about that fact. Same parts of the paper are compared to 2010 which is also quite obsolate data.

Response: Thank you for the opportunity to clarify this point. Actually, the data was collected between Oct/2014 to Mar/2016 (see the first paragraph of section 2.3). The first paragraph of section 3 had been reworded to eliminate this misunderstanding.

It is important to highlight that; it is a unique study in São Paulo city involving the main stakeholders, including many associations and supported by the municipality. To our knowledge, there is no such type of study regarding OHD so far. Indeed, we compare our results with Holguín-Veras et al. (2010), which is the closest to our approach. It should be noted that we are not comparing data at different times of the São Paulo OHD Pilot, neither using the same research protocols developed by other authors in more recent literature.

We added new information about current affairs at the end of the last paragraph. Thank you for this suggestion.

2) Some companies names are given and another are not. Authors should make some statement about that as well.

Response: Thank you for this remark. There is no reason to keep those names; we eliminated the name of the companies

3) Table 1 should be mentioned more - what is its reason and results, any objectives? It is not clear.

Response: Thank you for the opportunity to improve this part. We improved the explanation of this regulation before the table to be clearer. We also have added a new mention of the consequence of this scheme on page 8, just before section 3.1.1.

We keep this table because it is easy to see how circulating of cars works in São Paulo city, based on “scheme-license-plate-based car rotation.” An example of its importance comes just after the table.

We introduce in this section (2.2) the restrictions and regulations to trucks in São Paulo city, adopted by the municipality, and it is one of them which impact the delivery time. However, we have not measured this impact on the decision to deliver at night or during the daytime; neither we have measured the impact of other police on this decision.

4) Incorrect recording of hours should be improved: ”to run between 5 and 12h.”

Response: Done.

5) Acronyms such FLV and other should be described with its FIRST use.

Response: Thank you for this remark. Done throughout the paper.

6) ”We then generated scores that represent physical distances on the perceptual map” - how were these prepared? Authors should describe the methodology and give results not in figures’ way only. These should be taken into consideration for figures 1-3.

Response: Thank you for this remark. We have added a new section (2.4.1 Definition of the category coordinates (scores) in the perceptual map). Please, see it on the methodology part. It was also taken into consideration for Figures 1-3.

7) The kind of challenge mentioned in the paper is described in qualitative format. It needs quantitative one, therefore it is suggested to continue the research as such. Some method as FMEA, prediction markets, simulation modelling might be of sophisticated use.

Response: We use tables 3-5 to show this quantitatively.

8) Why is this reference at the end of the sentence? What is its meaning: ”As future work, specific areas inside São Paulo city could be chosen to develop OHD, taking account the opinion of main stakeholders: retailers, logistics operators, residents, and public administrations [27].” How does it support the meaning?

Response: We are planning to carry other research in specific parts of the city, for example, in places where there is a population with high/low income or areas with different density. We have taken the reference out. We also improved the sentence, thanks.

9) Since „The problems in parking during the day are unanimous among the drivers” it might be good to co-operate on this matter for future research as e.g. in  especially Kostrzewski M., Varjan P., 2018, The Issue of Parking Areas Conditions in Surrounding of Logistics and Production Facilities in Slovakia and Poland, Proceedings of 22nd International Scientific Conference. Transport Means 2018, Part II, October 03 – 05, 2018, Trakai, Lithuania, pp. 791-796 or Igarashia K., Okumura S., https://doi.org/10.1016/0886-7798(93)90138 --L

Response: Thank you for this suggestion. However, we could find this paper. The link does not work, and there is no access to these Proceedings. We also improved the text regarding future works.

10) Prediction markets might be used as well, based on Czwajda et al., http://doi.org/10.17270/J.LOG.2019.329

Response: Thank you for this suggestion. However, we found this reference a bit out of the scope. It is related to the “decision-making support instrument in vehicle recycling sector”. Then, we have not considered this reference.

11) ”It is a fact that there is a lack of structure for receiving goods, among the main problems are the excess of steps, narrow doors, the queue of trucks and the inadequate conditions of the receiving docks.” -  yes that’s right, therefore methods for facilities design might be used for future research. By the way, it is highly proposed not to use such generalised statements without any references.

Response: Thank you for this suggestion. We have added a reference for that.

12) Authors stated ”The long queues of trucks while waiting to unload merchandise is also cited as a favorable reason for delivery freight at night” however no citation is given. It should be specified.

Response: Thank you for this remark. Actually, it is the opinion of the drivers. We have specified this in the text.

13) Future results should be mentioned in the paper in more developed way. Aspects of urban logistics might be enriched with simulation studies as it was suggested before, e.g. proposed in: https://doi.org/10.1016/j.proeng.2017.06.077

Response: Thank you for this suggestion. It´s considered.

Reviewer 2 Report

Topic of the paper is suitable to the journal based on analysed problems (like noise etc.).

The paper has good structure, but I suggest to divide discussion part for two parts: discussion and conclusion.

Because Authors finished data collecting in March of 2016, they should also add some information about current state. There are a lot of actions taken in cities during last years related to sustainability development

Authors have to improve quality of the Figures 1, 2 and 3.

Author Response

Dear Reviewer,

Thank you for reviewing our article. Your comments have enabled us to improve our paper. The amended document accompanies this response. Please see our responses below for specific details on how we followed your comments and suggestions.

Topic of the paper is suitable to the journal based on analysed problems (like noise etc.).

1) The paper has good structure, but I suggest to divide discussion part for two parts: discussion and conclusion.

Response: Thank you for this suggestion. We have separated this. We have just followed the suggestion of the Sustainability template.

2) Because Authors finished data collecting in March of 2016, they should also add some information about current state. There are a lot of actions taken in cities during last years related to sustainability development.

Response: Thank you for this suggestion. We have added a new paragraph (see in the conclusion section) about OHD in São Paulo City. However, there is no news, due to the companies have been doing OHD independent of the local Government actions. We are preparing a second round of this research and adding some special variables related to cargo theft/security, technologies used, and also some investigating in different areas considering low/high population income and different density areas.

3) Authors have to improve quality of the Figures 1, 2 and 3.

Response: Thank you for this suggestion. It´s done.

Reviewer 3 Report

This study focuses on the analysis of logistics operators and retailers regarding freight deliveries in the city of São Paulo. A total of 100 logistics operators and 84 retailers participated in the survey, which was administered as a part of this study. A set of insights and relevant issues have been identified and discussed in the paper. In general, I think that the paper fits well into the scope of the journal. However, some revisions are required before the paper can be considered for publication. Certain segments of the paper must be strengthened. Below please find more specific comments:

*Abstract: The abstract of the paper can be improved. In particular, I recommend expanding the abstract and add a few sentences, highlighting the outcomes and importance of the conducted work.

*Page 1: The authors start the introduction section with a short discussion, highlighting complexity of logistic operations and product distribution. However, this discussion should be strengthened. In particular, the authors should clearly highlight the importance of freight transportation modes, international trade, and shipping as well as their critical role for the economic development of numerical countries. Also, the discussion should be strengthened by adding the following references, which highlight the importance of freight transportation and freight terminals.

Lalla-Ruiz, E., Expósito-Izquierdo, C., Melián-Batista, B., & Moreno-Vega, J. M. (2016). A set-partitioning-based model for the berth allocation problem under time-dependent limitations. European Journal of Operational Research, 250(3), 1001-1012.

Dulebenets, M.A., 2018. The vessel scheduling problem in a liner shipping route with heterogeneous fleet. International Journal of Civil Engineering, 16(1), pp.19-32.

Ursavas, E., & Zhu, S. X. (2016). Optimal policies for the berth allocation problem under stochastic nature. European Journal of Operational Research, 255(2), 380-387.

Zhen, L., Liang, Z., Zhuge, D., Lee, L. H., & Chew, E. P. (2017). Daily berth planning in a tidal port with channel flow control. Transportation Research Part B: Methodological, 106, 193-217.

Dulebenets, M.A., 2018. Green vessel scheduling in liner shipping: Modeling carbon dioxide emission costs in sea and at ports of call. International Journal of Transportation Science and Technology, 7(1), pp.26-44.

Xiang, X., Liu, C., & Miao, L. (2018). Reactive strategy for discrete berth allocation and quay crane assignment problems under uncertainty. Computers & Industrial Engineering, 126, 196-216.

Fonseca, G. B.; Nogueira, T. H.; Ravetti, M. G. A hybrid Lagrangian metaheuristic for the cross-docking flow shop scheduling problem. Eur. J. Oper. Res. 2019, 275(1), 139-154.

After this discussion, it would be logical to provide some statistics, showing the importance of logistic operations for the city of São Paulo specifically.

Page 3: At the end of the introduction section please add a paragraph, defining the structure of the manuscript (i.e., what the readers should expect in the following sections).

*Page 5: The authors collected the data from 84 stores, which belong to 17 different companies. Was this data sample sufficient to draw statistically significant conclusions?

*Page 8: The authors discuss some of the issues that were identified as a result of conducted survey, including lighting, noise, and public safety. Why did you focus on these particular issues? How about some other importance issues (e.g., congestion, accidents, severity of accidents, deterioration of products transported, etc.)?

*Page 10: The discussion section should be strengthened. The authors should clearly highlight limitations of this study and how they will be addressed in future research.

Author Response

Dear Reviewer,

Thank you for reviewing our article. Your comments have enabled us to improve our paper. The amended document accompanies this response. Please see our responses below for specific details on how we followed your comments and suggestions.

This study focuses on the analysis of logistics operators and retailers regarding freight deliveries in the city of São Paulo. A total of 100 logistics operators and 84 retailers participated in the survey, which was administered as a part of this study. A set of insights and relevant issues have been identified and discussed in the paper. In general, I think that the paper fits well into the scope of the journal. However, some revisions are required before the paper can be considered for publication. Certain segments of the paper must be strengthened. Below please find more specific comments:

*Abstract: The abstract of the paper can be improved. In particular, I recommend expanding the abstract and add a few sentences, highlighting the outcomes and importance of the conducted work.

Response: Thank you for this suggestion. It´s done.

*Page 1: The authors start the introduction section with a short discussion, highlighting complexity of logistic operations and product distribution. However, this discussion should be strengthened. In particular, the authors should clearly highlight the importance of freight transportation modes, international trade, and shipping as well as their critical role for the economic development of numerical countries. Also, the discussion should be strengthened by adding the following references, which highlight the importance of freight transportation and freight terminals.

Lalla-Ruiz, E., Expósito-Izquierdo, C., Melián-Batista, B., & Moreno-Vega, J. M. (2016). A set-partitioning-based model for the berth allocation problem under time-dependent limitations. European Journal of Operational Research, 250(3), 1001-1012.

Dulebenets, M.A., 2018. The vessel scheduling problem in a liner shipping route with heterogeneous fleet. International Journal of Civil Engineering, 16(1), pp.19-32.

Ursavas, E., & Zhu, S. X. (2016). Optimal policies for the berth allocation problem under stochastic nature. European Journal of Operational Research, 255(2), 380-387.

Zhen, L., Liang, Z., Zhuge, D., Lee, L. H., & Chew, E. P. (2017). Daily berth planning in a tidal port with channel flow control. Transportation Research Part B: Methodological, 106, 193-217.

Dulebenets, M.A., 2018. Green vessel scheduling in liner shipping: Modeling carbon dioxide emission costs in sea and at ports of call. International Journal of Transportation Science and Technology, 7(1), pp.26-44.

Xiang, X., Liu, C., & Miao, L. (2018). Reactive strategy for discrete berth allocation and quay crane assignment problems under uncertainty. Computers & Industrial Engineering, 126, 196-216.

Fonseca, G. B.; Nogueira, T. H.; Ravetti, M. G. A hybrid Lagrangian metaheuristic for the cross-docking flow shop scheduling problem. Eur. J. Oper. Res. 2019, 275(1), 139-154.

After this discussion, it would be logical to provide some statistics, showing the importance of logistic operations for the city of São Paulo specifically.

Response: Thank you for this suggestion. We partially agree with this statement because this paper focuses on OHD (within urban freight distribution context). Then, we minimize a broaden discussion related to the importance of freight transportation modes, international trade, and shipping because these are out of the scope of urban logistics. However, we have considered those references as much as we could in the introduction part. Some statistics have been added.

Page 3: At the end of the introduction section please add a paragraph, defining the structure of the manuscript (i.e., what the readers should expect in the following sections).

Response: Thank you for this suggestion. It´s done.

Page 5: The authors collected the data from 84 stores, which belong to 17 different companies. Was this data sample sufficient to draw statistically significant conclusions?

Response: Thank you for the opportunity to clarify this point. We believe that because we collected the data from the main stores, which were applied OHD in São Paulo city. We have followed most of their activities in the Pilot test, in which many sectors and associations have participated and given some support. Moreover, as stated in line 207 “Around 45% of these companies are part of the top five in their industrial sector”.

Page 8: The authors discuss some of the issues that were identified as a result of conducted survey, including lighting, noise, and public safety. Why did you focus on these particular issues? How about some other importance issues (e.g., congestion, accidents, severity of accidents, deterioration of products transported, etc.)?

Response: Thank you for this suggestion. We have focused on these issues because they were observed as issues during the Pilot Test. We agree that other variables could fit in this case as you suggested. We are going to consider those attributes next research, and we state this in future works on the paper.

Page 10: The discussion section should be strengthened. The authors should clearly highlight limitations of this study and how they will be addressed in future research.

Response: Thank you for this suggestion. We have improved this part, adding the research limitations in the conclusion part.

We do hope that all the modifications that we introduced in the paper are satisfactory. All of your comments and observations have been an important and critical resource in reshaping our manuscript. Thank you very much again for all of these valuable suggestions for improvement.

The authors

Round 2

Reviewer 1 Report

Authors are right about lack of proceedings of Transport Means 2018. If only there would be possibility of sending files in the system. The second paper of Japanese researchers can be found in the Internet.

In the paper Authors mentioned that data were gathered between October 2014 and March 2015, meanwhile in responses between October 2014 and March 2016. Which one is true?

In the case of ”(I X J)” notation for multiplication should be used: ×.

Notations given in the text and in the equations should be exactly the same.

Equation (1), (2) are incorrect -  there is no information as to what parameter the summation is. Similar question is about equations (7) and (8) with dot instead of multiplication notes.

Tables should not be divided into two or more pages.

Why the Authors think that prediction markets should not be useful for their future research?

Author Response

Thank you again for reviewing our article. Please see our responses below for specific details on how we followed your comments and suggestions.

Authors are right about lack of proceedings of Transport Means 2018. If only there would be possibility of sending files in the system. The second paper of Japanese researchers can be found in the Internet.

Response: We found the paper “The Issue of Parking Areas Conditions in Surrounding of Logistics and Production Facilities in Slovakia and Poland” published by Kostrzewski M., Varjan P. in the Proceedings of the 22nd International Scientific Conference. Transport Means 2018, Part II, October 03 – 05, 2018, Trakai, Lithuania, pp. 791-796 or Igarashia K., Okumura S. on the Internet as you mentioned. However, we respectfully disagree with this specific insert because it is a general problem (as these authors said at the beginning of their paper). If we were to consider this article, we should consider many other important articles as well.

In the paper Authors mentioned that data were gathered between October 2014 and March 2015, meanwhile in responses between October 2014 and March 2016. Which one is true?

Response: You are right, it is a bit confusing. We rewrote the first paragraph of Section 2 (lines 110-112). Actually, the survey ran between October 2014 and March 2016. It started during the Pilot test with 45 stores (see line 129). Now, we overlooked the fact of mentioning the sample in Section 2.3 (lines 206-207).

In the case of ”(I X J)” notation for multiplication should be used: ×.

Response: This change has been made.

Notations given in the text and in the equations should be exactly the same.

Response: This change has been made.

Equation (1), (2) are incorrect - there is no information as to what parameter the summation is. Similar question is about equations (7) and (8) with dot instead of multiplication notes.

Response: This change has been made. See red parts, please.

Tables should not be divided into two or more pages.

Response: This change has been made.

Why the Authors think that prediction markets should not be useful for their future research?

Response: Because it is not related to our paper/research field.

We hope that all the changes we made in the paper are satisfactory. All of your comments and observations have been an important and critical resource in reshaping our manuscript. Thank you very much again for all of these valuable suggestions for improvement.

Yours sincerely.

The authors

Reviewer 3 Report

The authors have adequately addressed my original comments. Presentation of the manuscript has been improved. Therefore, I recommend acceptance.

Author Response

We thank you again for your valuable review.

The paper has been proofread by a British native speaker to check the spelling and correct the grammatical mistakes.